# Low Incidence of Acute Antibody-Mediated Rejection after HLA Desensitization in Living Donor Kidney Transplant Recipients

**DOI:** 10.3390/life12121993

**Published:** 2022-11-29

**Authors:** Constantino Fernández Rivera, Catuxa Rodríguez Magariños, María Calvo Rodríguez, Tamara Ferreiro Hermida, Marta Blanco Pardo, Andrés López Muñiz, Sara Erráez Guerrero, Leticia García Gago, Ángel Alonso Hernández

**Affiliations:** Nephrology Service, A Coruña University Hospital Complex, 15006 A Coruña, Spain

**Keywords:** HLA desensitization, plasma exchange, immunoglobulins, rituximab, acute rejection

## Abstract

Desensitization allows the performance of human leukocyte antigen (HLA)-incompatible transplants. However, the incidence of acute rejection (AR) is high. This study aims to analyze the incidence of AR after transplantation with HLA-incompatible living donors in patients who underwent desensitization. Patients were immunosuppressed with tacrolimus, mycophenolic acid derivatives, and steroids after being desensitized with rituximab, plasma exchange, and/or immunoadsorption with specific cytomegalovirus immunoglobulins. A negative complement-dependent cytotoxicity or flow cytometry crossmatch and a donor-specific antibody titer < 1000 mean fluorescence intensity (MFI) were used to determine desensitization efficacy. A total of 36 patients underwent desensitization, and 27 (75%) were transplanted. After a follow-up of 58 ± 58 months (Min–Max: 0.13–169.5), five episodes of AR occurred: two antibody-mediated and three T-cell-mediated. No differences were found in baseline calculated panel-reactive antibodies (cPRA), class I or II MFI, number of antibodies, or Relative Intensity Scale (RIS) between AR and non-AR patients. Patients with antibody-mediated AR had higher cPRA (NS), MFI class I (*p* = 0.07) and class II (*p* = 0.006), and RIS (*p* = 0.01). The two patients with antibody-mediated AR and one patient with T-cell-mediated AR lost their grafts. In conclusion, the incidence of acute antibody-mediated rejection after desensitization was 7.4%, which occurred early post-transplantation in patients with high MFI and was associated with early graft loss.

## 1. Introduction

Kidney transplantation is the best therapeutic option for renal replacement therapy in patients with chronic renal failure [1]. Despite the considerable increase in transplants in recent years, an imbalance persists between available organs and patients on the waiting list [2]. A number of these patients are highly sensitized, and their transplantation options include joining specific lists, such as the Kidney Allocation System (KAS) [3,4] or the PATHI (a care program for hypersensitized patients) [5], undergoing desensitization protocols, or entering cross-transplant programs, with the latter having a low probability of obtaining a donor [6].

Desensitization is a process that consists of lowering the level of antibodies, thus allowing adequate preparation prior to transplantation. Different protocols allow living or deceased donor donations, and can be performed while on dialysis [7], at the time of donation [8], or post-transplantation [9]. The best results have been obtained in patients with a living donor and a pre-transplant initiation.

The drawback of this option is the high rate of acute rejection (AR)as a result of the recognition and confrontation against foreign agents by the HLA system, and especially the rate of antibody-mediated AR, which ranges widely between 0% and 61%, with the most frequent incidence being around 25–30% [10,11]. In recent years, new protocols have been introduced showing promising results, both in patients in whom previous desensitization was ineffective [12] and in the reduction of antibody-mediated AR [13]. The heterogeneity of protocols, their application, the definition of effective desensitization, and the methodology used explain such disparate results.

We present our center’s experience of using a desensitization protocol in patients receiving a kidney transplant from an HLA-incompatible living donor. The primary goal of this study was to evaluate the incidence of AR using the protocol and discuss the differences with respect to other existing strategies.

## 2. Materials and Methods

### 2.1. Patients

From January 1981 to June 2021, the A Coruña University Hospital, Spain, performed 3500 kidney transplants: 332 were from a living donor, of which 63 were performed with an ABO-incompatible donor and 27 with an HLA-incompatible donor. In addition, 7 transplants from the national crossover transplant system were performed.

All patients who received an HLA-incompatible living donor kidney transplant were included in the study. Patients previously accepted and signed the HLA-incompatible living donor kidney transplant (TRDV HLA-i) protocol after receiving information from the services involved in kidney transplantation.

All patients had a positive crossmatch against their donor by complement-mediated cytotoxicity, flow cytometry (FC) against T and B cells, or by Luminex solid-phase assay techniques (One Lambda, Thermo Fisher Scientific; Canoga Park, CA, USA).

### 2.2. Desensitization

The desensitization protocol consisted of the following steps: One month prior to transplantation, rituximab 375 mg/m² was administered. One week before starting plasmapheresis or immunoadsorption (PP/IA), tacrolimus, mycophenolate mofetil (MMF) or mycophenolate sodium (MFS), and prednisone were started. PP and/or non-specific IA (Therasorb) were performed initially on a daily basis (except Sundays). After each PP/IA session, the patients received specific anti-CMV immunoglobulin at a dose of 100 mg/kg (Cytotect Biotest Pharma GmbH, Germany). Patients undergoing renal replacement therapy underwent hemodialysis sessions every 48 h after the apheresis sessions or 4 exchange sessions per day on peritoneal dialysis. Once the apheresis sessions had started, the antibody levels were monitored, or a crossmatch was carried out. When the CDC or FC crossmatch was negative (sera treated with pronase) and donor-specific antibody levels reached a mean fluorescence intensity (MFI) < 1000, the patient was accepted for transplantation. Once transplanted, PP or IA and immunoglobulins were performed on days +3, +5, and +8 (Figure 1).

### 2.3. Immunosuppression

Induction was performed with basiliximab (20 mg on days 0 and +4) or thymoglobulin (1.125 mg/kg, adjusted to leukocyte and platelet count, not exceeding 5 mg/kg of accumulated dose). A dose of 250 mg of methylprednisolone was administered prior to the first dose of basiliximab, while 500 mg, 250 mg, and 125 mg doses were used during the first three days of thymoglobulin administration. Prior to the administration of thymoglobulin, acetaminophen, diphenhydramine, and actcortin were administered.

Immunosuppression began on the day of admission for apheresis with tacrolimus, mycophenolic acid derivatives, and prednisone. Tacrolimus levels were established at 10–12 ng/mL in the first week, with a progressive reduction to reach 7–8 ng/mL from the 6th month on. The initial dose of MMF was 2 g/day, followed by a reduction to 1 g/day after a month. The MFS dose was 1440 mg/day, which was reduced to 720 mg/day after a month. Prednisone was started at 20 mg/day and progressively reduced to 5 mg by the 6th month.

Eculizumab: A patient with aHUS due to a factor H mutation who had relapsed after his first transplant received eculizumab pre-transplant and subsequently continuously every 2 weeks, according to the usual protocol.

### 2.4. Infection Prophylaxis

All patients were given prophylaxis with cotrimoxazole for 6 months and itraconazole for 4–6 months. Patients with a CMV-positive donor who were CMV-negative received valganciclovir for 3–6 months (Figure 1).

The patient who received eculizumab received meningococcal vaccination (tetravalent and B) and prophylaxis with penicillin V, 200 mg every 12 h.

### 2.5. Follow-Up

After discharge, patients were monitored for renal function, tacrolimus levels, and CMV and BK virus status.

In the event of renal function deterioration, the patient underwent a biopsy and a determination of specific donor antibodies. No per-protocol biopsies or systematic controls of specific donor antibodies were performed during post-transplant follow-up.

Delay in graft function was defined as the need for dialysis in the first week.

For the diagnosis of biopsy rejection, the Banff classification criteria were used [14].

Steroids (3 boluses of methylprednisolone 500 mg for three consecutive days), PP or IA, intravenous immunoglobulin (IVIG), rituximab, and eculizumab (in refractory cases) were used for the treatment of antibody-mediated rejection.

Steroids (3 boluses of methylprednisolone 500 mg for three consecutive days) and thymoglobulin were used to treat T-cell-mediated rejection if there was no response to steroids.

### 2.6. Variables Studied

Efficacy of desensitization, delayed graft function, AR, renal and patient survival, renal function, PP/IA sessions, MFI versus classes I and II pre- and post-apheresis, calculated panel reactive antibodies (cPRA), number of antibodies against classes I and II, RIS (Relative Intensity Scale) as described by Jordan et al. [15], and infections (cytomegalovirus (CMV) and BK virus infection).

### 2.7. Statistics

Statistical analysis was performed using SPSS software (version 15.0.1, Chicago, IL, USA). Differences between groups were evaluated using Student’s *t* and ANOVA tests for normally distributed quantitative variables, or the Mann–Whitney test when the distribution was not normal. Chi square and Fisher’s tests were used for comparing qualitative variables’ distributions between groups. Survival was evaluated according to the Kaplan–Meier and log-rank tests. A multivariate study using Cox regression and ROC curves was applied for sensitivity and specificity analysis.

## 3. Results

This study included 36 patients (19 women), with a mean age of 45.9 ± 13.4 years. The relationships between donors and recipients were father/mother (9), husband/wife (15), brother/sister (8), child (1), and others (3).

At baseline, the patients presented several antibodies against class I or II specificities: 1.58 ± 0.76 (1–4), with MFI 7378 ± 4046 (900–14775) against class I and MFI 6487 ± 4149 (1044–15600) against class II. RIS was 8.1 ± 6.9 (2–25). Crossmatch by CDC was positive in 18 patients (50%) and crossmatch by FC in 9 (25%), and 9 (25%) patients only presented DSA using solid-phase techniques (Luminex, One Lambda) (Table 1).

After 8 ± 3 PP/IA sessions (4–15), 27 patients (75%) were transplanted, with the protocol being ineffective in 9. Of these 9, 8 had a previous crossmatch by CDC and 1 by DSA (*p* = 0.02).

MFI class I, class II, and RIS were predictors of desensitization efficacy in ROC curves, establishing a cut-off point of <10,968 MFI for class I, <7050 MFI for class II, and <12.5 points for RIS (AUC 0.962, 0.806, and 0.850, respectively; Table 2).

In the postoperative period, 14 patients presented with hematoma or hemorrhage, requiring reintervention in 7 (25.9%) and transfusion in 20 (71%). In 4 patients (14.8%), there was a delay in the initial graft function.

After a follow-up of 58.3 ± 58.5 months (0.13–169), 5 patients (21.5%) presented AR; 2 were antibody-mediated and 3 were T-cell-mediated. One patient diagnosed with antibody-mediated AR also presented data concordant with T-cell-mediated rejection (2A in Banff’s classification).

Patients who presented AR had a higher baseline class II MFI, a higher RIS, and a positive crossmatch by CDC, although without statistical significance. Only tacrolimus levels in the first month were significantly higher in patients without rejection (Table 3).

In patients diagnosed with antibody-mediated rejection, we evidenced a higher baseline cPRA (NS), MFI class I (*p* = 0.07), MFI class II (*p* = 0.006), and RIS (*p* = 0.02) (Table 4).

One patient developed AR by antibodies on the eighth day after transplantation following a surgical intervention that required re-anastomosis of the vascular suture due to initial poor perfusion with a new start of WIT. In the initial biopsy on the eighth day, he presented with intense glomerulitis (g3) and capillaritis (cpt3). In addition, there was a mild tubulointerstitial inflammatory infiltrate with tubulitis (t1), interstitial hemorrhage, and endotheliitis (2A in the Banff classification). He presented positive focal staining for C4d, although the patient was also ABO-incompatible. DSA titers at the time of biopsy after three post-transplant immunoadsorption sessions were less than 1000 MFI. Prior to desensitization, he presented an MFI of 9100 against class I antigens (B7 and B40) and an MFI of 15,600 against a class II antigen (DR 15). The patient did not respond to steroids, apheresis, or Ig, presenting histological data of rejection mediated by antibodies and, in the last biopsy, a pattern of thrombotic microangiopathy, leading to the start of eculizumab. A transplantectomy was performed, which showed evidence of AR and transplant glomerulopathy. It is possible that the rejection mediated by antibodies was not directed against HLA but against the monocytic endothelial system; however, anti-endothelial antibodies or other non-HLA markers were not evaluated.

One patient developed AR by antibodies a month and a half post-transplantation associated with obstructive uropathy due to ureteral stenosis and acute pyelonephritis. The biopsy showed evidence of glomerulitis (g3), capillaritis (cpt3), and positive C4d staining. No interstitial infiltrate, tubulitis, or endotheliitis were observed. Anti-HLA antibody titers had increased to 6400 for class I (B41) and 9500 for class II (DR13). She did not respond to steroids, apheresis, immunoglobulins, rituximab, or splenectomy, and ended up losing the graft.

One patient presented with T-cell-mediated rejection on the sixth day after transplantation, concomitantly with urinary sepsis that made it impossible to administer thymoglobulin and/or mycophenolic acid derivatives due to severe thrombopenia. He suffered a renal rupture after the biopsy due to edema secondary to inflammation. In this case, the determinations of DSA and C4d were negative. This patient also lost the graft.

Two patients presented with AR by T cells due to errors in taking medication in one case (six months) and poor adherence in the other (three years). Both had a good response to steroids.

Patient survival was 91.7% in the 5th year and 80.9% in the 10th year (Figure 2). The causes of death were sepsis of urinary origin, subarachnoid hemorrhage, acute cholecystitis, and multiple organ failure. Graft survival was 88.3% in the 5th year and 62.8% in the 10th year (Figure 2). Graft survival was worse in patients with AR, although without statistical significance. Causes of graft loss were AR (3), chronic kidney disease (4), recurrence of the underlying disease (2; aHUS and GN C3), and arterial thrombosis in another patient.

Graft survival was lower than in HLA-compatible patients (*p* = 0.03), although there were no differences in the rejection rate. HLA-incompatible kidney transplantation did not pose an increased risk for graft survival in the Cox analysis (Table 5).

Creatinine was 1.32 ± 0.78 mg/dL at one year, 1.38 ± 0.37 at the third year, and 1.68 ± 0.77 at the fifth year; proteinuria was 0.33 ± 0.34 g/24 h at one year, 0.25 ± 0.22 at the third year, and 0.43 ± 0.65 at the fifth year. Tacrolimus levels before kidney transplantation were 11.27 ng/mL, thereafter oscillating during the follow-up period, i.e., 8.12 ng/mL in the first year and 7.2 ng/mL in the fifth year.

There were eight CMV infections (29.6%), no BK viremia, and eight post-transplant neoplasms, including one case of chronic lymphatic leukemia in the third year, three epidermoid cutaneous carcinomas, one quiescent myeloma (underlying pathology), one cutaneous melanoma in the fifth year, one native kidney tumor in the fourteenth year, and one cervical carcinoma in situ.

## 4. Discussion

Due to the increasing number of hypersensitized patients on the waiting list and their high degree of sensitization, new desensitization protocols have been incorporated into clinical practice using complement inhibitors [13]; inhibitors of interleukin 6 or its receptor [16,17,18]; inhibitors of B cell [19] or plasma cell proliferation; or imlifidase [12]. Despite improving graft survival, the rejection rate continues to be high, specifically antibody-mediated rejection.

The incidence of AR varies from 0 to 60% in different series [20,21], although in recent years it has usually been between 15 and 30% [7,9,12,13,22]. Little is known about the mechanisms behind these numbers and regarding AR’s good response to treatment, which allows graft survival rates of 90% at five years.

In our series, the incidence of AR was 21%, with 7.4% (2) of it being mediated by antibodies. The rejections occurred on the eighth day (date of the biopsy) and 1.5 months after transplantation. In the first case, we believe that the trigger was the transplant surgery with arterial re-anastomosis, which added warm ischemia that could have been the trigger for the immune response. In this case, the absence of DSA and the presence of vascular involvement (endotheliitis) may lead to the consideration of another type of non-DSA- preformed antibody directed at the glomerular endothelium, as described by Delville et al. [23]. In the second case, the AR was related to an episode of obstructive uropathy and urinary sepsis.

Patients who presented AR measured by antibodies had a higher cPRA, a higher pre-transplant MFI, a higher number of antibodies class I and II, and therefore a higher RIS. In one patient, the pre-desensitization crossmatch had been by positive CDC, and in another by positive FC. Morath [24] using immunoadsorption with Globbafim columns and rituximab without immunoglobulins described that 8 out of 10 patients had MFI < 1000 by the time of transplantation, and found antibody-mediated AR in three patients who presented elevated MFI post-transplant class I and/or II (one at 1.5 months due to renal deterioration, another at 3 months in a protocol biopsy, and another with systemic lupus erythematosus presented thrombotic microangiopathy at the 6th month). This group published results in 2021 showing 38 patients with an incidence of acute rejection of 21%, with antibody-mediated acute rejection of 16% [25].

Rogers [20] did not present any rejection episodes in 10 patients who underwent a protocol similar to ours, even though 6 out of his 10 patients had DSA with MFI between 1178 and 12,141 at the time of transplantation. The rest were either negative or below 1000. Three patients were unable to desensitize, presenting MFI values between 8000 and 18,000. The Los Angeles protocols with IVIG or IVIG and rituximab required a negative CDC crossmatch [7], and the DSA titer was unknown at the time of transplantation. The antibody-mediated rejection rate was 22%. In Boston, Riella et al. [21], used a protocol similar to ours and induced with basiliximab, performed the transplant when the CDC crossmatch was negative even in the presence of DSA, although the titer was not mentioned. The incidence of rejection was 61%. In Vienna, Schwaiger et al. [8], used a protocol with immunoadsorption and a deceased donor immediately before transplantation and found an antibody-mediated rejection rate of 41%. At the time of transplantation. these patients presented higher levels of class I and II antibodies, MFI, and MFI sum (when there was more than one antibody). Amrouche et al. [9], using post-transplant desensitization and a deceased donor, found a rejection rate of 32.6% in the first year after transplantation (124 days post-transplantation) and a chronic rejection rate of 39.5%. At the time of transplantation, the patients presented DSA with MFI between 3800 and 5000, persisting at 60.9% at three months, but with lower MFI (1500–2800). In the first year, 50% of the patients still had DSA with MFI between 1280 and 2100.

In 2019, the effect of adding eculizumab [13] in patients previously subjected to a desensitization protocol was first described. At baseline, patients presented 2.7 (1–6) antibodies with a maximum MFI between 8000 and 8700 and an MFI sum between 15,000 and 17,000. After randomization, the eculizumab group had 11.8% antibody-mediated rejection compared to 29.4% in the non-eculizumab group. The immunological and clinical characteristics of the patients who presented with a rejection were not reported.

Results of using imlifidase, an enzyme derived from *Streptococcus pyogenes* that has the peculiarity of degrading and cleaving immunoglobulins, were reported by Jordan et al. [12]. The patients included in the study had a mean of 2.3 ± 8 HLA antibodies with MFI 5660 class I and 8199 class II. Acute antibody-mediated rejection happened in 10 out of 25 patients between 2 weeks and 5 months post-transplant. The immunosuppression received was different according to whether the group was from Sweden or Los Angeles, which may have modified the antibodies’ MFI at one month after transplantation. Recently, the efficacy of desensitization has been reported using inhibitors of interleukin 6 (clazakizumab) or its receptor (tocilizumab) (16–18) in addition to RXT, PPF, and immunoglobulins. The possibility of transplantation is very high since the desensitization criteria include the presence of negative CDC, even in the presence of antibodies from class I MFI < 5000 or class II MFI < 10,000. The incidence of AR-mediated by antibodies in the first year was 20–40% with tocilizumab and 20% when using clazakizumab, with 98.5% and 99.9% cPRA, respectively.

In conclusion, the rejection rate seems to be associated with the definition of desensitization, negative CDC, weak negative or positive FC, and DSA, being higher when patients start with greater pre- or post-desensitization levels. In our series and the Heidelberg series, DSA titers post-desensitization were below 1000 MFI and associated with a low antibody-mediated rejection rate, although that may be conditioned by not having performed protocol biopsies in our series. Rejection usually occurs early and is associated with a higher number of antibodies and a higher MFI intensity before desensitization.

## Figures and Tables

**Figure 1 life-12-01993-f001:**
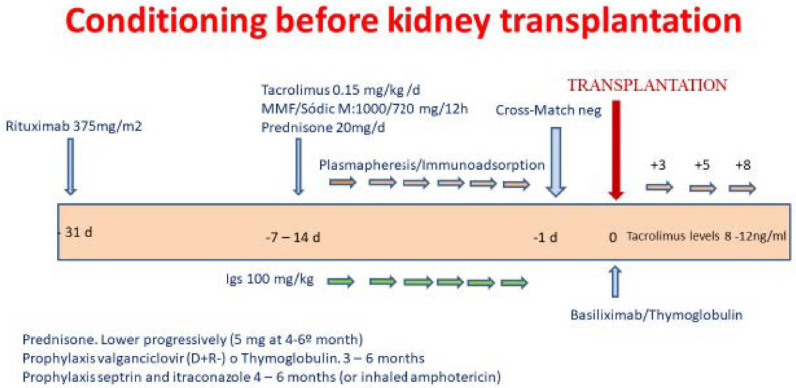
Desensitization protocol.

**Figure 2 life-12-01993-f002:**
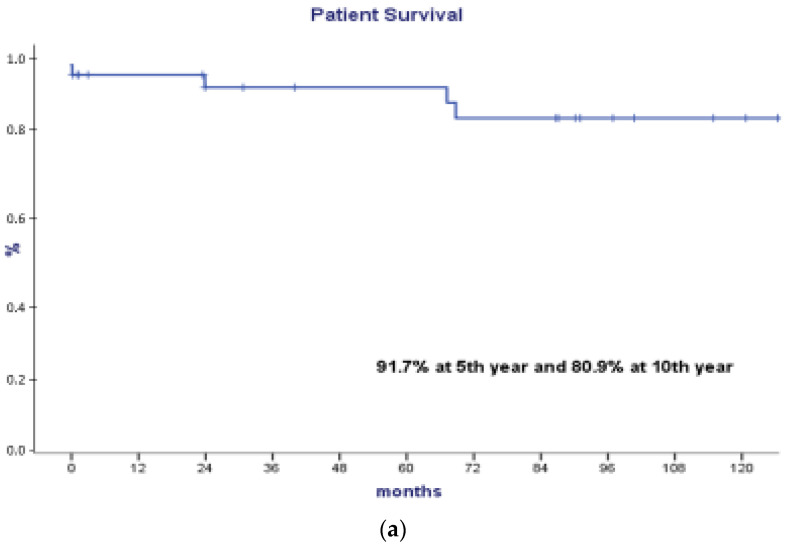
(**a**) Survival of patient. (**b**) Survival of graft.

**Table 1 life-12-01993-t001:** Baseline characteristics at the time of initiation of desensitization.

Variables	Mean ± SD (Min–Max)
Time on dialysis (months)	73.4 ± 78.8 (1–276)
Age (years)	45.9 ± 13.6 (21–71)
HLA (A, B, DR) incompatibilities	3.5 ± 1.2 (0–5)
cPRA (%)	77.4 ± 25.5 (5–100)
Number of HLA antibodies	1.6 ± 0.7 (1–4)
MFI class I	7748 ± 4046 (890–14775)
MFI class II	6487 ± 4149 (1044–15600)
RIS *	8.1 ± 6.9 (2–25)
	Number (frequency)
Gender:	
Male	17 (47%)
Female	19 (53%)
Underlying disease	
Not affiliated	7 (19.4%)
Polycystic disease	7 (19.4%)
Glomerulonephritis	6 (16.7%)
Nephroangiosclerosis	3 (8.3%)
Diabetes	1 (2.8%)
Tubulointerstitial nephritis	2 (5.6%)
Others	10 (27.8%)
Dialysis mode	
Hemodialysis	26 (69.4%)
Peritoneal dialysis	5 (13.9%)
Pre-dialysis	6 (16.7%)
Positive crossmatch	
CDC	18 (50%)
Flow cytometry	9 (25%)
Luminex	9 (25%)

* RIS: Relative Intensity Scale: <5000 MFI = 2; 5000–10,000 = 5; >10,000 = 10 for each antibody.

**Table 2 life-12-01993-t002:** Variables associated with the efficacy of desensitization.

	Efficacy Yes (n = 27)	Efficacy No (n = 9)	*p*
Age (years)	48.4 ± 13	38.6 ± 12	0.06
Donor age (years)	49.5 ± 11	49.3 ± 6	0.96
Time on dialysis (months)	86.8 ± 87	39.3 ± 46.5	0.16
Previous transplants			0.43
YES	14 (70%)	6 (30%)	
NO	13 (81%)	3 (19%)	
cPRA	78.7 ± 24	73.2 ± 26	0.58
MFI class I	5880 ± 2943	12,258 ± 2044	0.001
MFI class II	5701 ± 4029	9160 ± 3725	0.10
RIS *	5.8 ± 4.8	15 ± 7.9	0.01
Number of apheresis sessions	8.3 ± 2.7	8.7 ± 2.3	0.71
Positive crossmatch			0.02
CDC	10 (55%)	8 (45%)	
FC	9 (100%)	0	
Luminex	8 (89%)	1 (11%)	

* RIS: Relative Intensity Scale: <5000 MFI = 2; 5000–10,000 = 5; >10,000 = 10 for each antibody.

**Table 3 life-12-01993-t003:** Characteristics of patients depending on whether they presented acute rejection or not.

	Acute Rejection (n = 5)	No Acute Rejection (n = 22)	*p*
CPRA	75.4 ± 30	79.5 ± 24	0.74
MFI class I	5198 ± 3664	5714 ± 2791	0.74
MFI class II	8011 ± 7700	4990 ± 2111	0.49
Anti HLA * number	2 ± 0.7	1.4 ± 0.8	0.17
RIS **	9.6 ± 7.6	4.9 ± 3.7	0.24
HLA incompatibilities	3 ± 1.2	3.8 ± 1	0.14
Basal tacrolimus levels (ng/mL)	8.1 ± 2.8	11.9 ± 5.5	0.15
Tacrolimus levels at 1 month (ng/mL)	8.9 ± 3.9	18.5 ± 7.6	0.02
Number of apheresis sessions	8 ± 2.4	8.4 ± 2.8	0.74
Igs dose mg/kg (total)	800 ± 244	836 ± 235	0.79
Crossmatch			0.48
CDC	3 (60%)	7 (31.8%)	
FC	1 (20%)	8 (36.4%)	
Luminex	1 (20%)	7 (31.8%)	

* HLA: human leukocyte antigen. ** RIS: Relative Intensity Scale: <5000 MFI = 2; 5000–10,000 = 5; >10,000 = 10 for each antibody.

**Table 4 life-12-01993-t004:** Characteristicsaccording to present rejection mediated by antibodies or by T cells.

	Acute Antibody Rejection (n = 2)	Acute T-Cell Rejection (n = 3)	*p*
CPRA	98 ± 0.7	60 ± 30	0.18
MFI class I	8600 ± 707	2930 ± 2700	0.07
MFI class II	14,650 ± 1343	1372 ± 463	0.006
Anti HLA * number	2.5 ± 0.7	1.6 ± 0.5	0.23
RIS **	17.5 ± 3.5	4.3 ± 2.5	0.01
HLA incompatibilities	2.5 ± 2.1	3.3 ± 0.5	0.67
Basal tacrolimus levels (ng/mL)	9.2 ± 0.3	7.4 ± 3.8	0.56
Tacrolimus levels at 1 month (ng/mL)	8.3 ± 5.3	9.6 ± 3.8	0.80
Number of apheresis sessions	10 ± 2.8	6.6 ± 1.1	0.32
Igs dose mg/kg (total)	1000 ± 283	666 ± 115	0.32
Crossmatch			0.39
CDC	1 (50%)	2 (66.7%)	
FC	1 (50%)	0	
Luminex	0	1 (33.3%)	

* HLA: Human leukocyte antigen; ** RIS: Relative Intensity Scale: <5000 MFI = 2; 5000–10,000 = 5; >10,000 = 10 for each antibody.

**Table 5 life-12-01993-t005:** Cox analysis for graft loss.

Variable	Hazard Ratio	Confidence Interval 95%	*p*
Age (years)	0.90	0.83–0.93	0.02
Donor age (years)	1.03	0.95–1.12	0.41
Acute rejection	4.03	0.27–63.2	0.32
HLA * incompatible	5.05	0.44–57.6	0.19
Compatibilities	0.44	0.20–0.97	0.04
Time on dialysis	1.07	0.94–1.22	0.27

* HLA: human leukocyte antigen.

## Data Availability

Not applicable.

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
