# Peer review of "Low Incidence of Acute Antibody-Mediated Rejection after HLA Desensitization in Living Donor Kidney Transplant Recipients"

_life, 2022, doi:10.3390/life12121993_

Round 1

Reviewer 1 Report

This is a fine manuscript, but it is disappointing not to have in the first paragraphs a simple, layman's terms, definition of what Desensitization is and what is its purpose (though the authors late define "effective desensitization" which is a related but separate term). So please provide a simple straightforward definition for Desensitization. Similarly in the introduction define, in very simple terms, what is meant by acute rejection. Any non-specialized reader sent to Google search these terms is a reader lost unnecessarily. Everything else looks fine

Minor point: in line 53 "...Hospital, Spain, has performed..." - delete the "has"

Reviewer 2 Report

Suggestions:

1. to change "HLA incompatible transplants." to crossmatch positive transplants. 

2. Figure 3 can be removed

3. Figure 4 can be removed

4. Table 6 can be removed

5. Typographical errors line 44, 135, 290, 323, 326

6. Sentence construction line 274
